# Determinants of Maximum Magnetic Anomaly Detection Distance

**DOI:** 10.3390/s24124028

**Published:** 2024-06-20

**Authors:** Hangcheng Li, Jiaming Luo, Jiajun Zhang, Jing Li, Yi Zhang, Wenwei Zhang, Mingji Zhang

**Affiliations:** 1Sino-German College of Intelligent Manufacturing, Shenzhen Technology University, Shenzhen 518118, China; lihangcheng@sztu.edu.cn (H.L.); luojiaming299@163.com (J.L.); 202100403083@stumail.sztu.edu.cn (J.L.); 202300102040@stumail.sztu.edu.cn (Y.Z.); zhangwenwei@sztu.edu.cn (W.Z.); 2Sanechips Technology Co., Ltd., Shenzhen 518055, China; zhang.jiajun@sanechips.com.cn

**Keywords:** detection distance, magnetic anomaly detection, magnetic object, magnetic sensor

## Abstract

The maximum detection distance is usually the primary concern of magnetic anomaly detection (MAD). Intuition tells us that larger object size, stronger magnetization and finer measurement resolution guarantee a further detectable distance. However, the quantitative relationship between detection distance and the above determinants is seldom studied. In this work, unmanned aerial vehicle-based MAD field experiments are conducted on cargo vessels and NdFeB magnets as typical magnetic objects to give a set of visualized magnetic field flux density images. Isometric finite element models are established, calibrated and analyzed according to the experiment configuration. A maximum detectable distance map as a function of target size and measurement resolution is then obtained from parametric sweeping on an experimentally calibrated finite element analysis model. We find that the logarithm of detectable distance is positively proportional to the logarithm of object size while negatively proportional to the logarithm of resolution, within the ranges of 1 m~500 m and 1 pT~1 μT, respectively. A three-parameter empirical formula (namely distance-size-resolution logarithmic relationship) is firstly developed to determine the most economic sensor configuration for a given detection task, to estimate the maximum detection distance for a given magnetic sensor and object, or to evaluate minimum detectable object size at a given magnetic anomaly detection scenario.

## 1. Introduction

Magnetic Anomaly Detection (MAD) serves as a supplementary exploration technique alongside optical, radar, and thermal detection methods, due to its superiorities of being passive, rapid, and noninvasive, which are widely used in areas of Unexploded Ordnance (UXO) Detection [1,2,3], mineral exploration [4,5] and shipwrecks search [6,7]. Figure 1 shows the schematic of MAD in a shipwreck rescue. 

The detectable distance, defined as the maximum range at which a device can accurately identify a specific target, is the most important figure of merit for a given MAD system. Table 1 summarizes the technical determinants, namely object size, measurement resolution, and detectable distance of existing MAD platforms. Among them, measurement resolution is estimated based on published measurement results. We see from Table 1 that substantial efforts have been dedicated to extending the detectable distance by means of improving measurement resolution, background noise rejection ratio, and magnetization of magnetic markers.

### 1.1. Measurement Resolution

The measurement resolution of the MAD system, defined as the smallest increment or step size that a MAD system can detect or measure, is intrinsically determined by the sensitivity and electronic noise floor of the magnetic sensor [14].

Among magnetic sensors for MAD application, fluxgate is believed to be the most stable and reliable magnetic sensor for MAD application, but only capable of achieving an equivalent magnetic noise spectrum density of nT/$\sqrt{Hz}$-level due to the limitation of Barkhausen noise [15]. In seeking higher sensitivity, optically pumped magnetometers are adopted to give pT/$\sqrt{Hz}$-level equivalent magnetic noise spectrum density, which was tested to be effective in marine UXO detection and identification within a detection distance range of less than 20 m [9,16,17]. Thanks to the simultaneous advantages of high sensitivity and a low electric noise floor, SQUID pioneered an exceptionally low equivalent magnetic noise spectrum density of less than 10 fT/$\sqrt{Hz}$, ultimately offering pT-level measurement resolution to give a detection distance of 33 m for a UXO with a typical size of 2.1 m [18]. However, applications of optically pumped magnetometers and SQUID are limited because of their bulk size, massive loads and complex electronic accessories [19]. Besides, MAD engineers are delighted to see developments on novel magnetic field sensing mechanisms such as magnetoelectric, magneto-resistive sensors with integrated merits of miniature size, low electronic noise floor and high sensitivity [20,21,22].

### 1.2. Background Noise

The measurement resolution of the MAD system is extrinsically determined by background noises. On the other side of the coin, ongoing research on compensation algorithms, orthogonal basis functions (OBFs), and gradiometers have been conducted to improve the background noise rejection ratio, thereby achieving finer measurement resolution [10,23,24,25]. For instance, Prof. Phelps from the Minerals, Energy, and Geophysics Science Center in the U.S. derived a five-factors compensation model for UAV (Unmanned Aerial Vehicle) aeromagnetic system. This model successfully realized a noise rejection ratio up to 10 times at altitudes of 20~25 m, to give an overall measurement resolution of sub-nT level [25]. Approaches based on the decomposition of the MAD signal in the space of OBFs were also effective for background noise rejection and, therefore, greatly improved the resolution of the MAD system [26,27,28]. Gradiometer is another technique to reject background magnetic field fluctuations [28,29]. For example, Prof. V.G. Lucivero from Princeton University describes a 0.2 cm baseline magnetic gradiometer, to give a common mode noise rejection ratio up to 10^4^ and fT-level measurement resolution [30].

To recap by merging the intrinsic and extrinsic factors of magnetic sensors and background noise, respectively, better measurement resolution enables greater detectable distances. Regarding UXO detection, when the sizes of the UXOs are similar, Trammell [8] uses a fluxgate sensor with a resolution of 3 nT to detect a 0.2 m artillery shell, achieving a detectable distance of 7 m. Meanwhile, Mu [3] uses a magnetoelectric sensor with a resolution of 20 nT to detect the 0.3 m simulated UXO, with a detection distance of 2 m. Therefore, the overall measurement resolution of the MAD system is one of the dominant merits of detection distance.

### 1.3. Magnetization

For applications such as traffic and surgical magnetic makers, MAD is employed to identify the positions of these soft-magnetic or magnet markers [31,32,33,34]. These applications may allow for the artificial control of the magnetization strength of the markers to optimize the detection distance. The MAD system measures spontaneous magnetic fields generated from magnet markers [7,35], while disturbances of environmental magnetic fields due to flux concentration effects around soft-magnetic markers [3,6]. Practically neglecting permeability differences between most iron-made markers ($800<\mu_{r}<1200$), and remanent flux densities between most magnet marks (1.0 T $<B_{r}<1.6\;T$), the magnetization of objects is approximately proportional to the third power of the object size. In other words, larger objects result in a further detectable distance under the assumption of the dipole model [26]. As an illustration, we see from Table I that the MAD system with pT-level measurement resolution achieved a maximum detectable distance of 29 m for a ship with a size of 10 m [11], while giving a detectable distance limitation of 500 m for an under marine vehicle with the size of 100 m. Therefore, the object size is the other dominant factor of detection distance.

In summary, numerous previous researches provided detection distance as a technological metric of their MAD system. However, they fail to quantitatively interpret detectable distance dependency on the aforementioned factors, specifically detection resolution and target size.

To fulfill the purposes of identifying the most cost-effective magnetic field sensor for a given detection task, estimating the maximum detection range for a given magnetic sensor and target, and assessing the minimum detectable target size in a particular MAD scenario, it is necessary to establish an empirical formula of detecting distance by selecting measurement resolution and object size as principal variables.

In this paper, field MAD experiments are conducted on NdFeB magnets and a cargo ship based on a UAV platform, to calibrate finite element analysis (FEA) models of magnetic marker and soft-magnetic object, respectively. Measurement resolution and object size are parametrically investigated on calibrated FEA models, to give a “distance-size-resolution logarithmic” empirical formula.

## 2. Basic Principles

### 2.1. Magnetic Field Anomaly Generated by Object

Magnetic anomaly detection technique measures and interprets deviations or anomalies in the magnetic field around magnetic objects. By adopting the dipole assumption of the object, magnetic field anomaly $H_{a}$ generated by objects can be described as [26]:(1)$$H_{a}=\frac{3\left(m\cdot e_{r}\right)\cdot e_{r}-m}{4\pi {\left|r\right|}^{3}},e_{r}=\frac{r}{\left|r\right|},$$
where $r={\left(x,y,z\right)}^{T}$ is measure point position vector with respect to the dipole center, and m is the magnetic moment of the object, determined by its magnetization **M** and geometric size as:(2)m=∭MdV

This means that larger object size and stronger magnetization result in a higher magnetic moment, and eventually a higher abnormal magnetic field at a given position from (1), which also implies that the magnetic anomaly field diminishes with the third power of the distance from the target.

### 2.2. ***M**-**H*** Constitutive Relationships of Magnetic Object

Depending on the material properties, magnetic field anomaly can either arise from soft-magnetic material-made object or permanent magnet object, whose **M-H** constitutive relationships can be linearly or non-linearly described as:(3)$$M=(\mu_{r}-1)H_{b}$$
or
(4)$$M=\mu_{recoil}H_{b}+M_{r},$$
$\mu_{r}$, $\mu_{reciol}$ and $M_{r}$ in (3) and (4) denote relative permeability for soft magnetic material, recoil permeability and remanent magnetization for permanent magnet, respectively. $H_{b}$ is environment background magnetic field.

### 2.3. Governing Equations for FEA

For numerical analysis, an arbitrary magnetic object can be discredited into a finite number of tetrahedral elements with shape functions of cubic-order magnetic scalar potential, $\varphi_{m}$. Thereby, the superposition of magnetic field anomaly $H_{a}$ and background magnetic field $H_{b}$ can be expressed by the minus gradient of $\varphi_{m}$ as
(5)$$H=H_{a}+H_{b}=-\nabla \varphi_{m}.$$

Since MAD is usually conducted in an atmosphere environment, a linear **B-H** relationship is adopted in the measurement space as:(6)$$B=\mu_{0}\mu_{r}H.$$

Under the volumetric constraints of Gaussian-theorem ∇·B=0 and magnetic insulation boundary conditions at exterior infinite domain boundaries n·B=0, magnetic scalar potentials in (5) are solved using Galerkin method, to give the solution of overall magnetic fields.

### 2.4. Measurement Resolution, Sensor Equivalent Magnetic Noise Spectrum Density and Background Magnetic Field Noise Spectrum Density

Measurement resolution is inherently determined by sensor equivalent magnetic noise spectrum density and background magnetic field noise spectrum density over measurement bandwidth ∆f. The former is the key indicator of sensor signal quality raised from circuit noise, thermal noise (1/*f* noises) and vibrations. While, the latter is an important indicator to evaluate environmental noise fluctuations arising from the earth’s activity, solar activity and the existence of other magnetic objects. Both sensors have equivalent magnetic noise spectrum density $n_{s}$ and background magnetic field noise spectrum density $n_{b}$ can be quantified in units of $nT/\sqrt{Hz}$.

Therefore, the measurement resolution R at the interested frequency $f_{0}$ can be expressed as:(7)$$R=\sqrt{∆f}\cdot \max\left(\int_{f_{0}-\frac{∆f}{2}}^{f_{0}+\frac{∆f}{2}}n_{s},\;\int_{f_{0}-\frac{∆f}{2}}^{f_{0}+\frac{∆f}{2}}n_{b}\right).$$

For instance, the sensor noise was calibrated using a spectrum analyzer. At $f_{0}=1\;kHz$, with a spectrum analyzer bandwidth ∆f=100 Hz, the sensor equivalent magnetic noise spectrum density is $n_{s}=9\;nT/\sqrt{Hz}$. Also, in an actual test scenario, due to the magnetic field variations over space and time, the background magnetic field noise spectrum density is $n_{b}=5\;\mu T/\sqrt{Hz}$ at the aforementioned frequency and bandwidth. Therefore, it is evident that the measurement resolution is predominantly determined by the background noise spectrum density, $R=5\;\mu T/\sqrt{Hz}$.

## 3. Experiments

### 3.1. The MAD System

As shown in Figure 2, a MAD system was developed for the purpose of detecting magnetic fields emanating from ships and NdFeB magnet markers. The MAD systems consist of the UAV platform (M200, DJI^®^, Shenzhen, Guangdong, China), magnetic sensor, (HMC2003, Honeywell^®^, Charlotte, North Carolina, USA), positioning module, and communication module.

The curb weight, endurance time, and size are evaluated to be 6.7 kg, 30 min and 50 × 50 × 35 cm^3^, respectively.

Five three-axis GMR magnetic sensors are orthogonally assembled on UAV as shown in Figure 2 to eliminate the interference of the UAV to the sensors. It is featured to be wide range, high sensitivity, low equivalent magnetic noise spectrum density and low thermal drift of ±200 µT, 100 V/µT, $9\;nT/\sqrt{Hz}$ and −600 ppm/°C, respectively. These features facilitate the design of a compact sensor and enable ceaseless measurement capabilities. Figure 3 shows the calibrated sensor equivalent magnetic noise spectrum density of the MAD system using a spectrum analyzer (MDO34-SA1, Tektronix^®^, Beaverton, OR, USA).

To facilitate the conversion of magnetic field information into a digital stream for transmission across various modules, we employed a high-speed Analog-to-Digital Converter (ADC) with a sampling rate of up to 1 kHz. Furthermore, Real-Time Kinematic (RTK) technology was utilized for positioning purposes. The positioning module received data from both the positioning satellite and the ground reference station simultaneously. By employing the carrier phase difference algorithm, the module calculated real-time, high-precision positioning information for the mobile measurement platform. This approach enabled us to achieve a positioning accuracy of 0.1 m.

During measurement, the UAV is under the control of a specialist with a Civilian Unmanned Aircraft Pilot License to ensure safety and flight accuracy. The magnetic data from ADC and position data from the RTK are packaged by a micro process unit, and transmitted to the server via the Wi-Fi module using the Message Queuing Telemetry Transport (MQTT) protocol.

To observe real-time magnetic data and the distribution cloud map, a LabVIEW-based human-computer interaction interface module is developed on a laptop of the ground reference station. Due to the presence of digital filtering and wireless data check algorithms, the magnetic data and position data recording rate is optimized to be five samples/s.

Consequently, by adjusting the flight speed, the spatial sampling accuracy can be effectively modified to meet the desired requirements. In the following experiments, the drone’s speed ranged from 1 m/s to 3 m/s.

### 3.2. Cargo Ship as Soft-Magnetic Marine Object

As a typical soft-magnetic object in the ocean, a cargo ship was used to carry out MAD experiments. In order to conduct field experiments, a moored cargo ship was selected as a marine magnetic object at Yantian Port of Shenzhen, China, which is located at coordinates 114.2731° E and 22.5839° N. The ship was specified to have a length and width of 25 and 5 m, respectively, with a tonnage of 200 tons.

As shown in Figure 4, a Cartesian coordinate system is established for the convenience of discussion and analysis. The head direction of the ship, as indicated by the compass, was calibrated to be 53° south by west. Additionally, it is worth noting that the UAV’s flight direction aligned with the stern of the ship, meaning that east corresponded to 53° north.

As shown in Figure 5d, the MAD experiments were conducted by scanning *x-z* cross-section D1 and *y-z* cross-section D2 of the cargo ship following the programmed s-shape flight path. Furthermore, a straight track line L3 scan is conducted along the *z*-axis, starting from 2 m above the bow of the ship.

As shown in Figure 5a,b, the strength of **B**, is obtained by taking the square root mean of each **B** component. Black dots indicate the data sampling positions, while the rainbow maps of **|B|** are obtained by 3D cubic spline interpolation to give a smooth visible result.

We see from cross-section D1 that there are two large areas of |B|, whose center points are 12.5 m and 30 m, respectively, among which the one at 12.5 m is caused by the stern, while the one at 30 m is caused by the bow. It can be obviously observed that the |B| of the bow is larger than the stern, which matches the actual situation that the volume of the bow is larger than the stern.

Magnetic abnormal signals in D1, D2 and L3 provide solid evidence that as the distance between the magnetic sensor and the cargo ship increases, there is a sharp drop of |B| until it reaches the geomagnetic noise floor. Furthermore, it has been observed that the distribution of |B| exhibits regular fluctuations, which could potentially be attributed to background magnetic field noise spectrum density from geomagnetic.

It is noteworthy that D1, D2, and L3 exhibit distinct variations in the background magnetic field noise spectrum density. This discrepancy could potentially arise from measurement errors resulting from the time interval between measurements spanning several hours. We see from Figure 5a–c that the minimum |**B**| level varies from 30.1 μT to 45.1 μT, which may be attributed to the power level of the UAV platform and the rotation of the earth over a long experiment duration. Under DC magnetic field conditions, the effect of frequency can be ignored. As shown in Figure 5c, after the magnetic field stabilizes, there is a background noise of 4.13 μT, which is greater than the sensor noise of 9 nT. Therefore, according to Equation (7), the measurement resolution under cargo ship conditions is 4.13 μT.

### 3.3. NdFeB Magnet as Magnetic Marker Object

The NdFeB magnet, renowned for its exceptional magnetic properties, usually serves as a prominent candidate for magnetic marking techniques.

In this study, a sandwich-structured self-made magnet consisting of six NdFeB magnets separated by wood plated with a gap distance of 10 mm is selected to represent a typical magnetic marker for sea reclamation settlement calibration application. Each axial magnetized NdFeB magnet is measured to give a diameter of 100 mm and thickness of 20 mm with a maximum magnetic energy product of 35 MGOe.

In comparison to Cargo Ship, the NdFeB magnet MAD experiment is conducted at the center of the sand stadium at Shenzhen Technology University which is located at coordinates 114.2367° E, 22.4240° N. As shown in Figure 6, a Cartesian coordinate system is established for the convenience of discussion and analysis. Additionally, the UAV’s flight direction is aligned with the y-axis, meaning 20 degrees North by West.

Magnetic signature scanning was carried out on the *x-z* cross-section (D1) and *y-z* cross-section (D2) planes above the magnet and the line (L3) parallel to the *z*-axis above the magnet, as shown in Figure 7d.

The experimental results are depicted in Figure 7, providing a visual representation of the measured data. Notably, the presence of the NdFeB magnet induces a noticeable magnetic anomaly signal. The observed behavior in this region follows a similar pattern to that illustrated in Figure 5. However, it is important to highlight that the measurement resolution of the system utilized in this particular example is estimated to be approximately 3.86 μT in Figure 7c.

## 4. Numerical Model

Two finite element models are established strictly following experiment configurations to investigate the magnetic signature distributions of cargo ships and magnet markers. Both models are developed based on the ‘magnetic field no current’ (MFNC) module in COMSOL Multiphysics^®^ version 5.5, and studied using a stationary solver under the assumption of reduced field analysis. The constitutive equations of the model are defined by Equations (2)–(6). Since certain parameters are unmeasurable, a calibration approach is adopted to adjust the FEA model based on experimental results. The geometries, background earth magnetic field, and material properties are defined with respect to the two experiment conditions to give consistent cross-sectional magnetic field maps according to the UAV flight path.

### 4.1. Finite Element Analysis of Cargo Ship MAD

As shown in Figure 8, the FEA geometry consists of a cargo ship (magnetic object domain), air domain, and infinite domain. Among which, based on the test, a cargo ship model of about 20 × 5 × 8 m^3^ is constructed in a 1:1 ratio; the air domain, a sphere with a diameter of 100 m, is an analogy to experimentally measurement space; the infinite domain is defined to concern the magnetic field attenuation to infinite distance.

In addition, the background Earth magnetic field of the test site needs to be defined. By entering the measured latitude and longitude coordinates into the International Geomagnetic Reference Field (IGRF) database, the local Cartesian coordinate-based magnetic field is obtained, $H_{bx}=-22,985\;nT,{\;H}_{by}=0.30369\;nT,{\;H}_{bz}=-0.24384\;nT$, which is then entered into the reduced field interface of the MFNC module. As defined in (3), an important determinant of the cargo ship magnetization is relative permeability, which is defined to be 1000 as a typical value of DH36 steel.

Figure 9 shows a magnetic field simulation of a cargo ship. Among them, Figure 9d is the sampling plane generated according to the flight trajectory of the UAV in the test. Figure 9a, Figure 9b, and Figure 9c are simulation results of *x-z* section D1, *y-z* section D2, and longitudinal intersection line L3, respectively.

Clearly, the results in Figure 9 show agreement with the experimental results in Figure 5. According to Figure 9a, the peak |**B**| is shown above both the bow and stern, and the value of the bow is greater than that of the stern. According to Figure 9b, the |**B**| decays cubed with the distance from the ship.

Nevertheless, due to the absence of background magnetic field noise spectrum density in the simulation and the differences in sampling for the three cases in the test, the minimum |**B**| in Figure 9 tends to be consistent at about 39.3 μT.

### 4.2. Finite Element Analysis of Magnet Marker MAD

In addition, for the simulation of hard magnetic target NdFeB, the magnetic target domain of geometric definition is built according to the magnets in the test, which is a cylinder with a diameter of 0.1 m and a height of 0.17 m, and the other two domains are consistent with those in Figure 8. The geomagnetic field where NdFeB Magnet is located is defined as $H_{bx}=-22,985\;nT,{\;H}_{by}=0.30369\;nT,{\;H}_{bz}=-0.24384\;nT$.

Figure 10 shows the simulation results of the magnet. Similarly, it also indicates the same rule as Figure 7, reflecting the consistency of simulation and test. The error between simulation and measurement is about 10%.

In sum, numerical model results are inherently consistent with experimental results. This proves that numerical models are successfully calibrated by experimental results. Moreover, the numerical model can fully predict the distribution of magnetic anomalies, which is helpful to the study of detectable distances.

## 5. Determination of Detection Distance

On the basis of the experimentally calibrated cargo-ship FEA model, the Maximum magnetic abnormal detection distance (*D*) dependence on object size (*S*) and measurement resolution (*R*) are investigated using parametric sweep on iso-metric scaling factor to analogy object size variation within the ranges of 1 m~500 m. The maximum magnetic abnormal detection distances (*D*) data are determined from spatial |**B**| distributions by extracting the minimum distance between *R*-iso-surfaces to object position.

Figure 11 shows the FEA results of the size-resolution determinant on maximum detection distance. We see from the log-log view of Figure 11 that ${log}_{10}$(*D*) is negatively linear proportional to ${log}_{10}$(*R*), which means, for a given *S*, a 1000 times increase in *R* results in an approximately 10 times increase of *D*.

In contrast, ${log}_{10}$(*D*) is positively linear proportional to ${log}_{10}$(*S*), which means, for a given *R*, a 10 times increase in *S* leads to approximately 7 times increase of *D*.

Based on the aforementioned findings, a linear relationship of ${log}_{10}$(*D*) influenced by ${log}_{10}$(*R*) and ${log}_{10}$(*S*) was established. The subsequent distance-size-resolution logarithmic relationship was fitted using a linear regression equation.
(8)$$-7.30*\log\left(S\right)+2.81*\log\left(R\right)+8.26*log(D)-7.82=0$$

Based on the distance-size-resolution logarithmic relationship formula described by (8), the analytical results of the size-resolution determinant on maximum detection distance are shown in Figure 12. The relative error between Figure 12 and Figure 11 is evaluated on average to be 1.14%. The maximum detecting distance, measurement resolution, and object size of existing MAD platforms are marked and illustrated in Figure 12. This analysis aligns with the distance-size-resolution logarithmic relationship described in (8).

In summary, the distance-size-resolution logarithmic relationship is established through the utilization of the calibrated FEA model. Subsequently, a dependable analytical relationship is derived by comparing it with existing platforms.

## 6. Conclusions

Aiming at quantitively illuminating the determinants of the maximum magnetic anomaly detection distance in MAD, field experiments have been conducted on soft and hard magnetic objects based on a UAV magnetic exploration platform.

Constitutive relationships for typical magnetic objects have been derived for MAD FEA modeling. Such constitutive relationships may theoretically elucidate the two mechanisms of magnetic anomalies raised from soft and hard magnetic objects, which is proven to be effective in FEA modeling to give coherent magnetic anomaly patterns between experiment and numerical results.

A distance-size-resolution logarithmic relationship is first discovered using an experimentally calibrated finite element model and empirically verified using literature data. Such discovery may be helpful to determine the most economical sensor configuration for a given detection task, estimate the maximum detection distance for a given magnetic sensor and object, or evaluate the minimum detectable object size in each magnetic anomaly detection scenario.

## Figures and Tables

**Figure 1 sensors-24-04028-f001:**
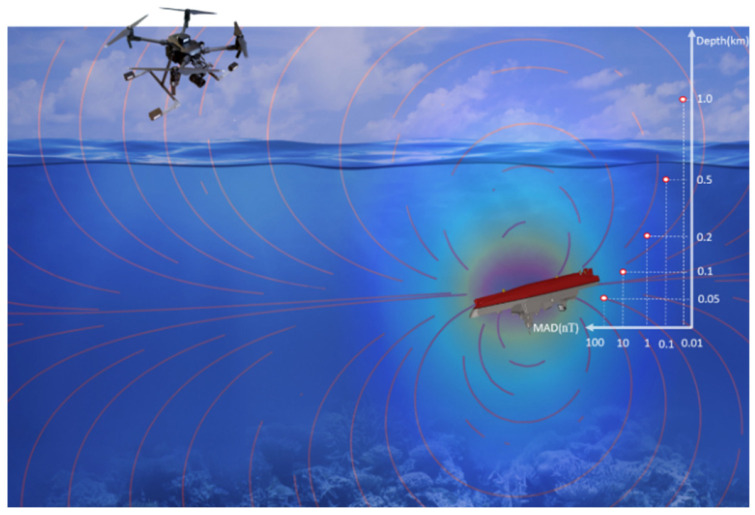
Unmanned Aerial Vehicle magnetic anomaly detection schematic for shipwreck rescue.

**Figure 2 sensors-24-04028-f002:**
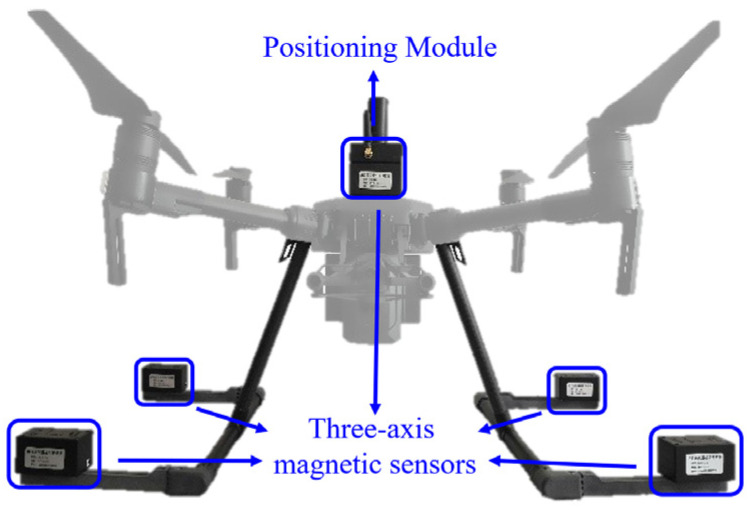
Magnetic abnormal detection system based on Unmanned Aerial Vehicle.

**Figure 3 sensors-24-04028-f003:**
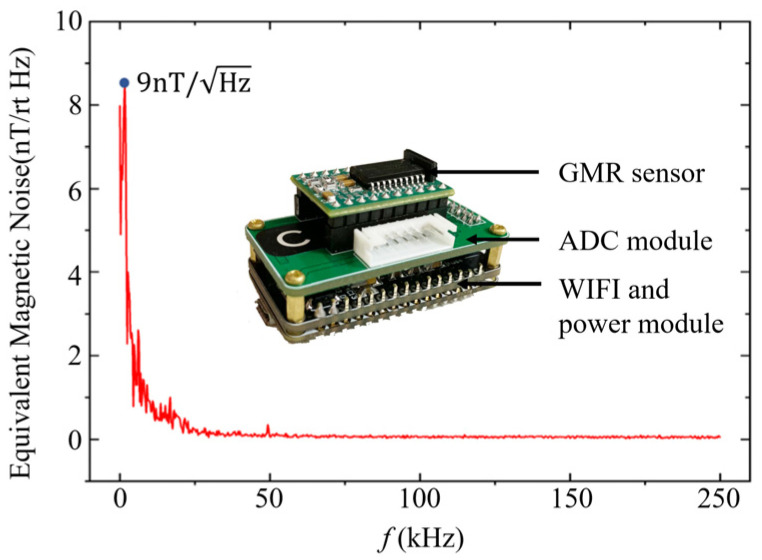
Equivalent magnetic noise spectrum density of magnetic sensor.

**Figure 4 sensors-24-04028-f004:**
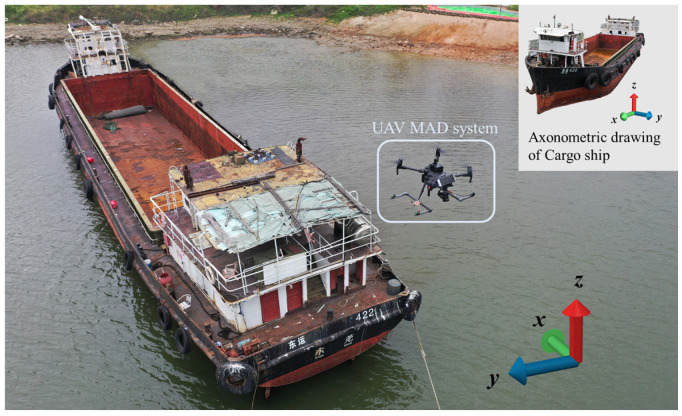
Cargo ship physical diagram.

**Figure 5 sensors-24-04028-f005:**
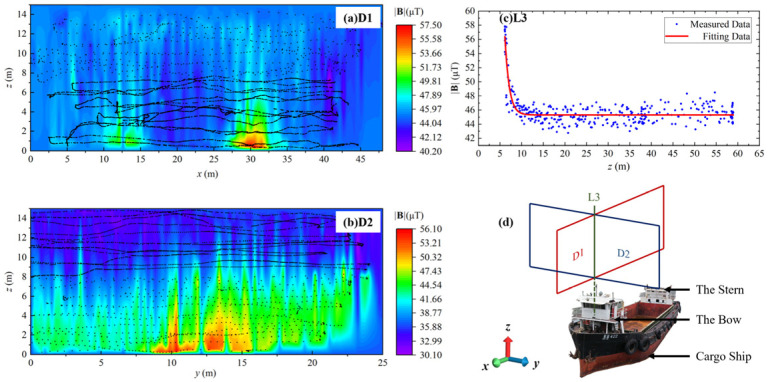
The measured results of |**B**| distribution of cargo ship, where black dots are flight track and data collection points: (**a**) D1 is the *x-z* cross-section of cargo ship, (**b**) D2 is the *y-z* cross-section of cargo ship, (**c**) L3 is the altitude intersection line of cross section, and (**d**) Diagrammatic sketch of cargo ship MAD measurement.

**Figure 6 sensors-24-04028-f006:**
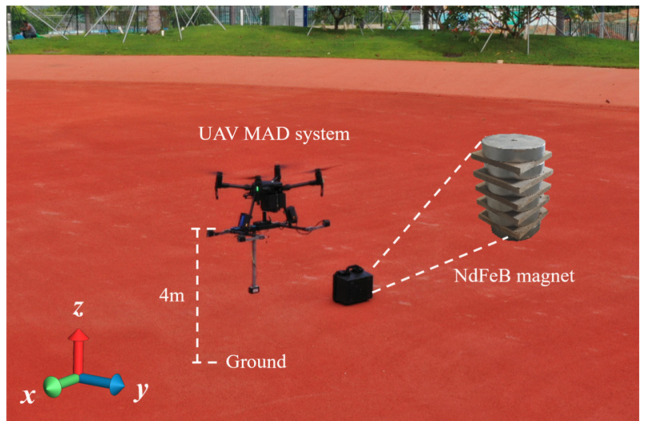
NdFeB magnet physical diagram.

**Figure 7 sensors-24-04028-f007:**
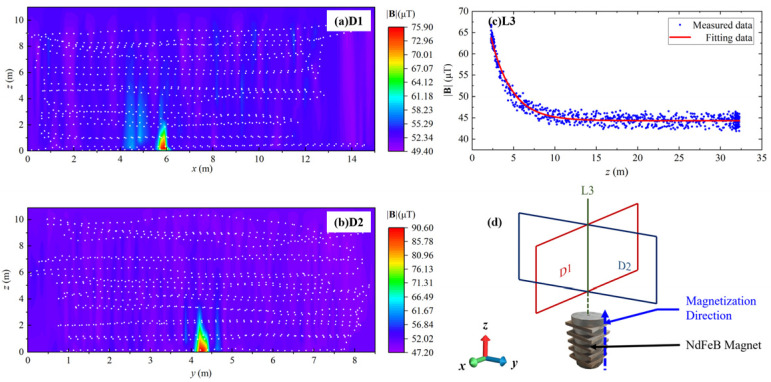
The measured results of magnetic field distribution of sandwich structured self-made NdFeB magnet, in which the white dots indicate flight tracks and data collection points: (**a**) D1 and (**b**) D2 are the cross-sectional magnetic field distribution of the magnet, (**c**) L3 shows the variation of magnetic field intensity along altitude over the magnet, and (**d**) Diagrammatic sketch of cargo ship MAD measurement.

**Figure 8 sensors-24-04028-f008:**
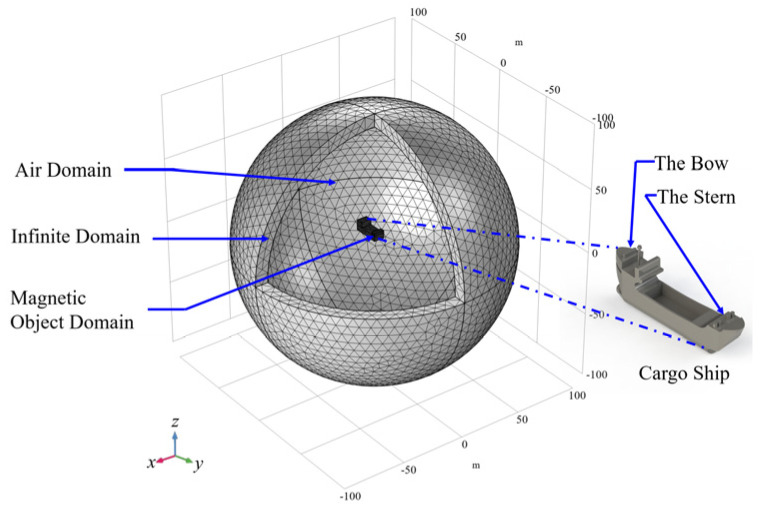
FEA simulation model mesh generation and Geometric model of cargo ship.

**Figure 9 sensors-24-04028-f009:**
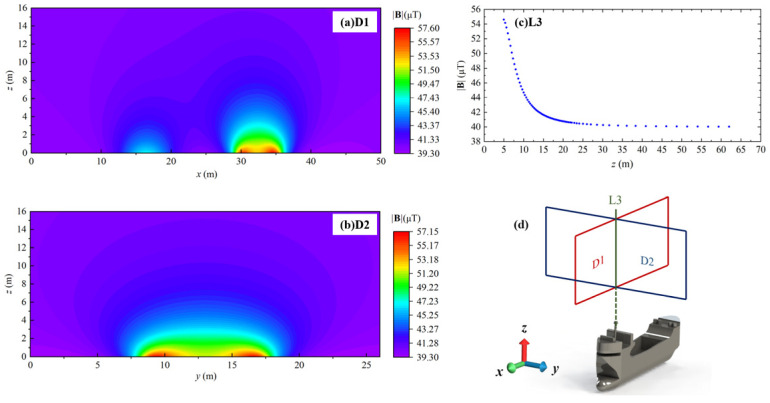
The simulated results of |**B**| distribution of cargo ship: (**a**) D1 is the *x*-*z* cross-section of cargo ship, (**b**) D2 is the *y*-*z* cross-section of cargo ship, (**c**) L3 is the altitude intersection line of cross-section, and (**d**) Diagrammatic sketch of cargo ship MAD simulation.

**Figure 10 sensors-24-04028-f010:**
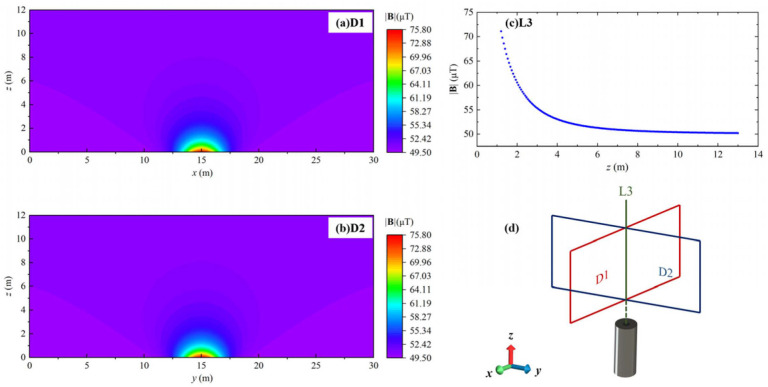
The simulated results of |**B**| distribution of NdFeB magnet: (**a**) D1 is the *x-z* cross-section, (**b**) D2 is the *y-z* cross-section, (**c**) L3 is the altitude intersection line of cross-section, and (**d**) Diagrammatic sketch of magnet MAD simulation.

**Figure 11 sensors-24-04028-f011:**
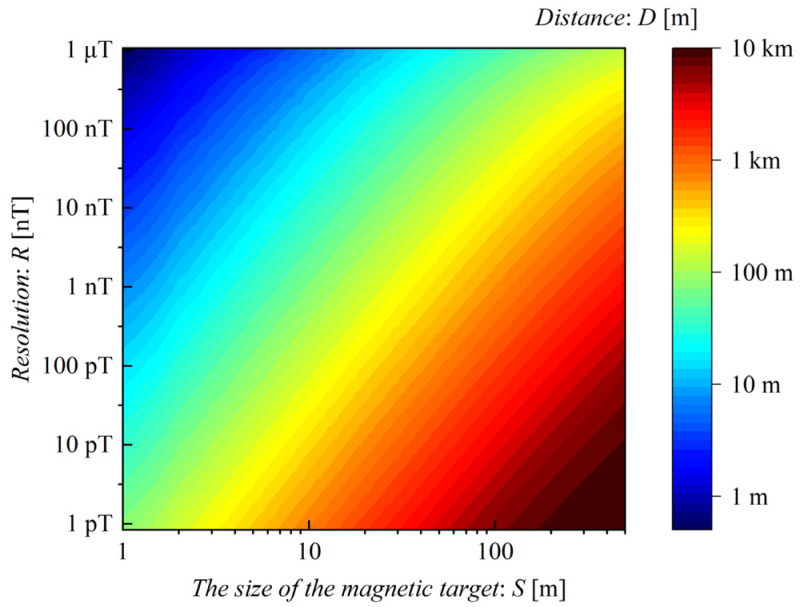
Experimentally Calibrated FEA Results of Size-Resolution Determinant on Maximum Detection Distance.

**Figure 12 sensors-24-04028-f012:**
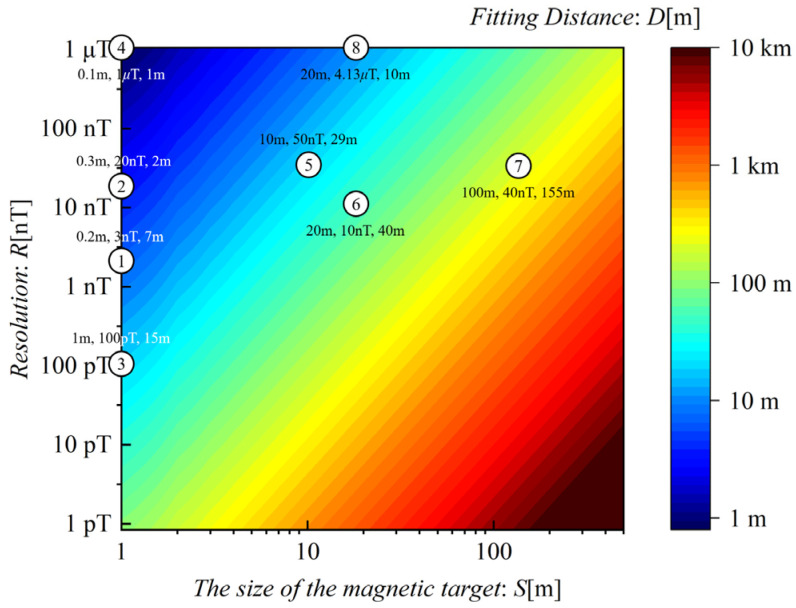
Analytical Results of Size-Resolution Determinant on Maximum Detection Distance. The No. corresponding to the platforms in Table 1 are enclosed within circles, with the adjacent label indicating the platform’s *S*, *R*, and *D*, respectively.

**Table 1 sensors-24-04028-t001:** Primary technical merits of existing MAD platforms.

No. Sensors (Noise Density)	Object (Size)	Measurement Resolution	Detectable Distance
1. Fluxgate ($0.1\;\mathrm{nT}/\sqrt{Hz}$)	Land UXO (0.2 m)	3 nT	7 m@2005 [8]
2. Magneto-electric ($2.5\;\mathrm{nT}/\sqrt{Hz}$)	UXO Simulation (0.3 m)	20 nT	2 m@2021 [3]
3. Optically pumped ($1\;\mathrm{pT}/\sqrt{Hz}$)	Land UXO (1 m)	100 pT	15 m@2022 [9]
4. Fluxgate ($1\;\mathrm{nT}/\sqrt{Hz}$)	UXO Simulation (0.1 m)	1 μT	1 m@2020 [10]
5. Magneto-electric ($6\;\mathrm{pT}/\sqrt{Hz}$)	Ship (≈10 m)	50 nT	29 m@2022 [11]
6. Electro-magnetic ($1\;\mathrm{pT}/\sqrt{Hz}$)	Gold Deposit (20 m)	10 nT	40 m@2021 [12]
7. Electro-magnetic ($1\;\mathrm{pT}/\sqrt{Hz}$)	Gold Deposit (100 m)	40 nT	155 m@2018 [13]
8. GMR ($9\;\mathrm{nT}/\sqrt{Hz}$)	Cargo Ship (20 m)	4.13 μT	10 m@This Work

## Data Availability

The data presented in this study are available on request from the corresponding author due to confidentiality of national funding.

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
