# Peer review of "Determinants of Maximum Magnetic Anomaly Detection Distance"

_sensors, 2024, doi:10.3390/s24124028_

Round 1

Reviewer 1 Report

Comments and Suggestions for Authors

Determinants of Maximum Magnetic Anomaly Detection Distance

The authors present an interesting measurement of the magnetic field anomaly generated by two target objects, a cargo ship and an permanent magnet array. A through review of the existing literature is presented in a coherent way. Field measurements were carried out with a UAV (drone) and analysis of the results are presented in terms of field resolution. This is supported by numerical simulation.

The main issue is there are many small and more significant issues with the English used throughout the manuscript. These needs to be addressed before publication.

Equation 7 is not well defined or explained. Additionally, the values for resolution for both the cargo ship and the permanent magnet are stated, but calculations are not shown. At least once this calculation should be given in full, with values for each parameter given, e.g. what are the values for n_s and n_b in both cases? What is \Delta f? These could be given in a table. It is not clear how Fig5.(c) and Fig7 (c) are used in this calculation.

Small note: Fig4. There are strange shadows around the stern of cargo ship, on the left hand side of the image. Is this a real photo or has it been edited in some way? If the images is a composition (inserting of drone) this should be stated.

Comments on the Quality of English Language

The main issue is there are many small and more significant issues with the English used throughout the manuscript. These needs to be addressed before publication.

Author Response

Dear Reviewer,

Thank you very much for your valuable feedback and insightful comments on our manuscript. We have carefully considered and adopted your suggestions. Detailed responses to your comments are provided in the accompanying PDF document. All changes made to the original text have been highlighted in blue for your convenience.

We appreciate your time and effort in reviewing our work.

Best regards,
Dr. Mingji Zhang

Reviewer 2 Report

Comments and Suggestions for Authors

The acronym OBFs is defined twice at lines 67-74. In this work, a magnetic anomaly detection (MAD) system is developed to detect magnetic fields emanating from ships and NdFeB magnet marker. Field MAD experiments are conducted on NdFeB magnets and a cargo ship based on an unmanned aerial vehicle (UAV) platform to calibrate finite element analysis (FEA) models of magnetic marker and soft-magnetic object. Measurement resolution and object size are parametrically investigated on calibrated FEA models to give a distance-size-resolution logarithmic empirical formula. Two finite element models are established following experiments configurations when investigating the magnetic signature distributions of cargo ship and magnet marker. Both models are developed based on magnetic field no current (MFNC) module in COMSOL Multiphysics, using stationary solver when assuming reduced field analysis. The numerical models results are consistent with the experimental results, showing that the numerical models are successfully calibrated by experimentation. A distance-size-resolution logarithmic relationship is established via a calibrated FEA model empirically verified using literature data. Authors claim that such discovery may be helpful to determine the most economical sensor configuration for a given detection task, to estimate the maximum detection distance for a given magnetic sensor and object, or to evaluate the minimum detectable object size in each magnetic anomaly detection scenario. Authors do not give any trace about how the constitutive relationships from the basic principles section are employed or implemented.

Author Response

(The authors gave the same response as above.)
